# Fast Fabrication of Complex Surficial Micro-Features Using Sequential Lithography and Jet Electrochemical Machining

**DOI:** 10.3390/mi11100948

**Published:** 2020-10-20

**Authors:** Ming Wu, Krishna Kumar Saxena, Zhongning Guo, Jun Qian, Dominiek Reynaerts

**Affiliations:** 1College of Mechanical and Electrical Engineering, Guangdong University of Technology, Guangzhou 510006, China; mingwucn@outlook.com; 2State Key Laboratory of Precision Electronic Manufacturing Technology and Equipment, Guangdong University of Technology, Guangzhou 510006, China; 3Department of Mechanical Engineering, KU Leuven and Member Flanders Make, 3001 Leuven, Belgium; krishna.saxena@kuleuven.be (K.K.S.); jun.qian@kuleuven.be (J.Q.); dominiek.reynaerts@kuleuven.be (D.R.)

**Keywords:** electrochemical micromachining, surface microstructures, micro letters

## Abstract

This paper presents fabrication of complex surficial micro-features employing a cross-innovative hybrid process inspired from lithography and Jet-ECM. The process is referred here as mask electrolyte jet machining (MEJM). MEJM is a non-contact machining process which combines high resolution of lithography and greater flexibility of Jet-ECM. It is a non-contact process which can fabricate variety of microstructures on difficult-to-machine materials without need of expensive tooling. The presented work demonstrates the process performance of this technology by statistical analysis and multivariate kernel density estimation (KDE) based on probabilistic density function. Micro-letters are fabricated as an example of complex surficial structure comprising of multiple intersecting, straight and curved grooves. The processing response is characterized in terms of geometrical size, similarity ratio, and cumulative shape deviation. Experimental results demonstrated that micro letters with good repeatability (minimum SD of shape error ratio 0.297%) and shape accuracy (minimum shape error of 0.039%) can be fabricated with this technology. The results suggest MEJM could be a promising technology for batch manufacturing of surface microstructures with high productivity.

## 1. Introduction

Electrochemical micromachining (ECMM) is a promising technology for fabricating surface structures [1]. In contrast to chemical etching (CE) which uses caustic acids/alkalis as a medium, ECMM uses a neutral saline solution [2]. The electrolyte usually has high conductivity and low toxicity with a potential to recycle electrolytes by employing ion exchangers. Thus, ECMM can be an ecofriendly alternative to CE [3]. The material removing rate in the ECMM can be adjusted by controlling the applied current. A wide range of hard yet conductive materials such as bulk metallic glasses [4], titanium alloys [5], superalloys [6], carbide-metals [7,8] can be effectively machined by ECMM. Several configurations of ECMM process such as jet electrochemical machining [9], scanning micro-electrochemical flow cell based ECMM [10], wire electrochemical micromachining [11], a tool-based hybrid laser-ECM [12] and through-mask electrochemical micro-machining [13] have been introduced to generate surface microstructures.

Jet electrochemical machining (Jet-ECM) uses an electrolyte jet which flows through the nozzle and hits the workpiece directly to localize the electrochemical reaction [14]. With this process; micro-dimples, micro-grooves and micro-slits can be easily fabricated [15]. Reference [16] demonstrated that the horizontal orientation of the jet is beneficial for the Jet-ECM processes to improve the machining accuracy. Reference [17] introduced the air-shielding in the Jet-ECM process and disclosed a novel scheme to improve the machining accuracy by a combination of theoretical analysis and experimental verification. Although, Jet-ECM is a promising method for creating surface structures but suffers from stray machining and less repeatability due to jet-dynamics, hydraulic-jump and requires tool-path planning for complex surface structures as in sequential machining process. Reference [18] used textured tools in jet-ECM consisting of arranged holes to fabricate microstructures resulting in a more efficient parallel machining process.

In recent years, several ECMM based new processes have been introduced such as Scanning micro-electrochemical flow cell based micromachining which has capability to fabricate micro-dimples and at the same time confining electrolyte to a droplet [10]. Similarly, a tool-based hybrid laser-electrochemical micromachining process proposed in Reference [12] which has capability to fabricate hierarchical microcavities in single-step.

Through mask electrochemical micro-machining (TMEMM) is hybrid method which employs photolithography to produce micro-patterns on photoresist coated on workpiece surface and ECMM to dissolve the areas exposed to the electrolyte. Thus, a large number of desirable areas dissolve in parallel. This technique is a precise and relatively fast process, capable of generating well defined surface textures with controlled size, location, even on a cylindrical inner surface [19], and density. With this method, Reference [20] fabricated three-dimensional cylindrical microstructures with feature sizes as small as 40 μm. Reference [21] presented a new development in the TMEMM of titanium using a laser-patterned oxide film. Reference [22] develop a modified TMEMM technique to fabricate micro-dimple arrays employing polydimethysiloxane mask which can be re-used. In TMEMM the electrolyte flow direction is normal to the patterned photoresist. The electrolyte flow will form a vortex, and has a low flow velocity in the mask hole [23]. The electrolytic byproducts and Joule heat must be transported away by the electrolyte flow to guarantee process accuracy and stability [24]. The electrolyte crossing the mask from one side to another can lead to a non-uniform flow field which results in poor consistency of material removal rate, and thus poor machining accuracy in surface microstructure fabrication [25]. Reference [26] reported that a mask with cone-shaped holes is beneficial for the electrolyte flow. Reference [13] proposed a modified forward electrolyte flow mode with a multi-slit structured cathode for good distribution of electrolyte flow field. All of the aforementioned methods are indeed very useful and inventive but require sophisticated equipment designed for structuring with predefined scales or shapes and require further research for real applications.

In the present work, a cross-innovative hybrid process coupling both the Jet-ECM and lithography is put forward. MEJM connects the strong points of lithography, which is a high-resolution process; and the flexibility of Jet-ECM, which has an adaptable flow-field. It avoids other limitations such as the non-uniformity of the TMEMM process, the inherent difficulty of manufacturing and assembling jet nozzles at micro-scale, and the inflexibility of designing micro-patterns for the corresponding jet nozzles. MEJM can solve this non-uniformity by employing nozzle travel. As shown in Table 1, for micro dimples fabrication, compare with TMEMM, MEJM [25] demonstrated a better consistency of the dimensional variation by the ratio of the standard deviation to the width of micro dimples.

In this paper, systematic experiments on the fabrication of complex surficial features by MEJM are carried out. Micro-letters are fabricated to demonstrate the potential of micro-fabrication by MEJM for large areas with repeating complex geometries. The multivariate kernel density estimation (KDE) based on probabilistic density function are employed to characterize the processing pattern from multivariate distribution. The results suggest that MEJM could be a promising technology for fast and batch manufacturing of surface microstructures.

## 2. Materials and Method

### 2.1. Description of the Method

(Figure 1 illustrates a schematic diagram of the MEJM process. The machining steps are explained below:a.Clean and spin coatingThe photoresist is spin coated on the workpiece surface which is cleaned in alcohol and acetone in an ultrasonic bath (Figure 1a);b.ExposureAfter the soft baking, a UV oven is employed to expose the photoresist through a photo mask (Figure 1b);c.Hard bakingThe patterned photoresist is structured on the workpiece surface through resist development and hard baking (Figure 1c);d.Mask electrolyte jet machiningThe electrochemical cell consisting of workpiece with patterned photoresist (anode) and an electrolyte nozzle (cathode) is activated. The electrolyte nozzle travels back and forth over the workpiece surface with a given velocity and in a certain crossing path, and electrochemical reactions and material dissolutions take place selectively (Figure 1d);e.micro lettersAfter the photoresist removal process, the result is micro letters which are fabricated over the workpiece within the exposed path (Figure 1e).

### 2.2. Experimental System

Figure 2 shows the experimental system which consists of a three axis motion system that provides movements to the electrolyte jet nozzle and workpiece. The traveling nozzle can provide consistent and continuous material removal within a large area of the workpiece, meanwhile ensuring the uniformity of the machining process. An electrical power and controlling system provides power supply to the workpiece and electrolyte jet nozzle during the machining process. Finally, a controlled electrolyte jet system is implemented using an electrolyte delivery system, and a pressure relief valve to supply fresh electrolyte at a high as well as controllable pressure through the nozzle. This ensures supply of fresh ions for electrochemical reactions and pushes away by-products of the machining process.

Table 2 lists the chemical composition of the stainless steel used as workpiece in the present study. Furthermore, a sodium nitrate electrolyte solution is employed in this experiment for good passive film formation which implies a uniform oxide film over the workpiece and limits stray dissolution. In this experiment, a positive photoresist (Shipley^®^ 1818) is employed because of its good compatibility and strong adherence with the metallic workpiece and its high lithographic resolution.

### 2.3. Experimental Design and Data Collection

MEJM is an ECMM based processing technique, thus applied voltage U and processing time are key parameters, which influence the precision and progress rate of micro-fabrication. In MEJM, a faster nozzle travel speed corresponds to a slower processing time, and vice versa. In order to realize the potential in micro-fabrication for large area repeating complex surficial geometries, an investigation on fabrication of micro-letters was carried out. The experimental conditions are listed in Table 3. The processing parameter window is based on author’s experimental experience from previous work [25].

The morphology of the patterned photoresist and of the fabricated workpiece was characterized by a confocal laser scanning microscope (Olympus^®^ 2000, can get the 3D geometry and section area at depth of interest) and a scanning electron microscope (Supra^®^ 55VP, can get the profile and elemental composition of target features).

## 3. Results and Discussions

### 3.1. Fabrication of Surface Micro-Letters

Figure 3 shows the exemplar surface micro-letters fabricated using MEJM technology. The photolithographic process is used to prepare the patterned photoresist, consisting of 112 micro letters, 16 (8×2) sets of 7 letters; then micro-letters were fabricated within every travelling path of the electrolyte jet. The minimum line width of micro letters is around 40 μm. As electrolyte nozzle is moving along the designed path, the surface electric current density on the workpiece varies as given in Figure 4a${}_{1}$–a${}_{4}$ and Figure 5d,e. Figure 4 depicts the simulated shapes using moving mesh feature in Comsol^®^ software 5.4 and the simulated shapes resemble very well to the experimental shapes as in Figure 5a. It can also be detected from Figure 4 and Figure 5a that the bottom surface of the engraved letters is not always flat. This can be explained from the current density distribution plots of the cross-sectional area as shown in Figure 5. It follows that the current density is also not uniformly distributed throughout the letter and hence this is reflected in the shape as well.

To be specific, the primary cause which contributes to the shape non-uniformity throughout the letter as seen from Figure 5d,e is that the current density along with the edges of the photoresist is always higher, and can be described by time-dependent simulations of electric current density. Another important reason for shape non-uniformity is the geometry shape. As shown in Figure 4a${}_{1}$–a${}_{4}$, it is evident that the electric current density is higher at the ends and at the parts with changing curvature. The current density distribution will be influenced by their geometry shape as well as the border letter engrave width can result in lower current density and lower material removal amount subsequently, as shown in Figure 5d,e.

To some extent, both of these reasons can be classified as one, the shaper geometry of the position, the stronger electric field indicated. It is sharper along the boundary of the photoresist and workpiece than it at the bottom. And, for the fabrication of micro letters, narrow grooves are more “sharp” than wide grooves at the macro level. The current density along with the edges of the photoresist are always higher, thus will eventually lead to an uneven bottom surface of the engraved letters.

Figure 4 depicts the simulated shapes using moving mesh feature in Comsol^®^ software 5.4 and the simulated shapes resemble very well to the experimental shapes as in Figure 5a. This explains that the non-uniformity of fabricated shapes corresponded to non-uniformity of the current density distribution throughout the letter.

### 3.2. Investigation on Shape Accuracy

The electrochemical reactions responsible for metal removal are isotropic in nature, and hence there is inevitably some undercutting below the photoresist in MEJM process. In conventional electrochemical material removal process, the index of machining localization is the over-cutting. The evaluation index of shape accuracy in Reference [27], such as diameter and depth to evaluate simple microstructures cannot be applied to complex geometries; so a more appropriate evaluation of machining shaping accuracy is required.

As shown in Figure 6, the features obtained from the MEJM are proportional to the photolithographic features, the similarity ratio ($\zeta_{s}$) is defined as in Equation (Equation 1):(1)$$\zeta_{s}=\frac{S_{t}}{S_{0}},$$
where $S_{0}$ is the area of the pattern prepared on the photoresist, and $S_{t}$ is the area of the pattern fabricated on the workpiece.

The shape deviation ΔS represents the area difference between the designed and machined pattern and can be calculated as follows:(2)$$\Delta S=|S_{t}-S_{0}|.$$

The shape error $\zeta_{e}$ which represents the deviation from the designed area can be obtained as follows:(3)$$\zeta_{e}=\sum \frac{\Delta S}{S_{0}}.$$

Typical patterned photoresist and corresponding micro features of micro letters are shown in Figure 3. Table 4 list the descriptive statistical analysis at different applied voltage *U* with $v_{1}=800\;\mu$m/s, 2400μm/s, and 4800μm/s, respectively. The data and figure indicate that the ratio of cumulative shape deviation $\zeta_{e}$ (mean value ranging from 1.478% to 17.70%) and machining depth *h* (mean value ranging from 0.293 μm to 7.775 μm) increased with increasing applied voltage *U* and decreasing nozzle travel speed *v*.

For a better understanding of MEJM process, the multivariate kernel density estimation (KDE) based on probabilistic density function described in Reference [28], as given in Equation (Equation 4) is used to investigate the processing pattern [29] in MEJM. It estimates the distribution of all the data observed so far, providing a statistical and intuitive method to illustrate the data panoramically [30]. The KDE plots help to characterize the processing pattern from multivariate distribution, which seeks to model the probability distribution of data points.

The kernel density estimate of *f* at the point *x* is given by
(4)$$\hat{f}\left(x,x_{i};t\right)=\frac{1}{n}\sum_{i=1}^{n}K\left(x,x_{i};t\right),$$
where
(5)$$K\left(x,x_{i};t\right)=\frac{1}{\sqrt{2\pi t}}e^{-\frac{{\left(x-x_{i}\right)}^{2}}{2t}}$$
is a Gaussian kernel, and $\sqrt{t}$ is referred to as the bandwidth, which influences the smoothness of the density reconstruction and has been intensively investigated on optimizing the bandwidth [31,32]. In this study, self-adaptive bandwidth was referred from Scott method [33].

Multivariate kernel density estimation of machine depth *h* and similarity ratio $\zeta_{s}$, as a non-parametric approach; the form of the density function is derived from the measurement data collection without any previously assumed specific distribution. Figure 7 gives a much more intuitive and accurate idea of the shape of the data distribution where a Gaussian distribution contributed at the location of each input point is employed based on the data by probability density function (pdf).

As an instance, the probability density value (PDV) of machining depth *h* and similarity ratio $\zeta_{s}$ at applied voltage U=50 V and nozzle speed $v_{1}$ = 800 μm/s are presented in Figure 7b,c respectively. The distribution plots show that the PDV (probability density value) of machining depth peaked at 0.7 μm and resembles a normal distribution. The PDV of similarity ratio peaked at 1.1029. The plots of PDV vs machine depth and similarity ratio are merged into a joint-description, as shown as Figure 7a. Normal distributions of PDV of machine depth and similarity ratio are illustrated in all the joint-descriptions, as shown in Figure 7a,d,e.

Since, the processing parameters stayed unchanged at all values of applied voltage and nozzle moving velocity, the machining depth *h* and similarity ratio $\zeta_{s}$ should theoretically remain the same, which represented a centralized distribution in terms of *h* and $\zeta_{s}$. Each of the measurements should be centred around a fixed value, the peakedness (kurtosis) value, at a given processing parameter if only measurement errors were assuming. However, as shown in Figure 7 and Figure 8, the data points presented a decentralized distribution.

Compared with nozzle moving velocity, the applied voltage address more decentralized character of which machining depth and similarity ratio distribution, and the similarity ratio experienced a slight but more perceptible decentralization within the processing parameter window. In Figure 7c, the distribution of similarity ratio even displayed a clear bimodal distribution.

The more normal distribution the measurements fit, the less decentralization and stochastic should be presented. This decentralization and stochastic distribution can be explained as measurement error partially, but the observable, implicit and organized variety of decentralization request further explanations. The observed variability can be explained based on the following two reasons:

1.intergranular corrosionIn case of stainless steel (depending on heat treatment), the susceptibility to intergranular corrosion can lead to non-uniform dissolution thereby deteriorating shape uniformity as illustrated in Figure 9. This phenomenon can be steady decline by pulse power supply [34] and will be investigated in future works.2.mask failureThe thin masks are prone to deformation, failure and delamination due to electrolyte flow field and machining conditions as illustrated in Figure 10. This can lead to increased overcut and deterioration of shape accuracy. This issue can be conveniently addressed by employing a thicker mask.

As shown in Figure 8, the distribution of machining depth *h* is much more centralized with applied voltage U=50 V, then this centralization decreased significantly. According to hypothesizes formed above, this significant decrease can be caused by both the intergranular corrosion and mask failure. Higher applied voltage leads to higher current density, which results in more intergranular corrosion and mask failure. Slower nozzle speed denotes more processing time and this can lead to a possibility of intergranular corrosion and mask failure which suggested a similar decentralized trend.

However, the decentralized trend for machining depth and the similarity ratio is quite different.

As shown in Figure 11 and Figure 12, the probability density value of the similarity ratio decreased less significantly (−53.915%, from 48.571 to 22.384), than that of the machining depth (−78.915%, from 2.433 to 0.513) while using a slower nozzle speed $v_{1}=800$μm/s.

Since the centralization of data distribution can be decayed by mask failure and intergranular corrosion, the probability density value can be regarded as an inverse indicator of a processing error in the electrochemical machining process. If there is a consensus that the isotropic material removal rate is one of the intrinsic features in ECMM, the intergranular corrosion grown approximately. Therefore, a fair deduction can be made that the main reason that probability density value of the similarity ratio decreased less significantly at a slower nozzle speed $v_{1}$ = 800 μm/s, is the possibility of mask failure increases dramatically at a low nozzle speed. This can be perceived from comparing the probability density value for applied voltage U=50 V in Figure 12a–c that when nozzle speed decreases from 2400 μm/s to 800 μm/s, the value of similarity ratio decreased by 71.917%, from 172.956 to 48.571. Whereas the value of machining depth decreased by 47.790%, from 4.66 to 2.433.

In a processing parameter window where higher nozzle speed was adopted, from to 2400 μm/s, the decreasing rate between similarity ratio (decreased by 18.991%, from 213.501 to 172.956) and machining depth (decreased by 14.101%, from 5.425 to 4.66) was very close.

Slower nozzle speed denotes longer processing time. For the mask, this increases the chance of exposure in the stirred and mixed flow field caused by the moving nozzle. As a consequence, it drives towards more opportunistic mask failure. The probability density value of similarity ratio and machining depth both influenced by intergranular corrosion. The value of the similarity ratio is also susceptible to mask failure. The results suggest that a fast nozzle speed is beneficial to a higher shape accuracy, which is consistent with the previous study [25].

Overall, the experiment results demonstrated a good machining accuracy (minimum shape error of 0.039%) in fabrication of complex geometries.

### 3.3. Investigation on Repeatability

The repeatability of the fabrication of micro letters can be indicated by the standard deviation (SD):(6)$$SD=\sqrt{\frac{1}{N-1}\sum_{i=1}^{N}{\left(x_{i}-\bar{x}\right)}^{2}}$$
where
(7)$$\bar{x}=\frac{1}{N}\sum_{i=1}^{N}x_{i}.$$

It can be observed from Table 5 that as the applied voltage *U* is increased or the nozzle travel speed *v* is decreased, the SD of the similarity ratio $\zeta_{s}$ (mean value ranging from 0.297 to 2.008), and depth *h* (mean value ranging from 0.080 to 0.747) is increased.

These results suggest that a faster nozzle translation would decrease the variation of similarity ratio $\zeta_{s}$ and improve the dimensional consistency.

As explained in Reference [25], the flow condition in TMEMM is unchangeable and hence it is difficult to discharge reaction products such as gas from the inter-mask space, hence by-products accumulate on the downstream side (Figure 13b${}_{1}$–b${}_{3}$). The asymmetry of by-products accumulation also happens in MEJM (Figure 13a${}_{1}$–a${}_{3}$), but with a travelling nozzle the electrolyte flow is mixed and stirred. By-products do accumulate on the downstream side, but to a lesser extent than in TMEMM because of the changes in flow lead to changes in the side of accumulation. The asymmetry of by-products accumulation contributes to the asymmetry of electrochemical reactions over the workpiece, and eventually leads to a dimensional asymmetry (Figure 13c).

Compared with TMEMM, MEJM exhibits a better performance in terms of dimensional variation. The SD of MJEM is smaller (minimum SD of shape error ratio s is 0.297%, which the ratio of the SD to the width of TMEMM in Reference [13] is 0.925%, and the ratio of the SD to the depth is 11.078%).

## 4. Conclusions

For conventional jet electrochemical machining (Jet-ECM), the minimal feature size depends on the inner diameter of the jet nozzle, so point-to-point processing of fine features will be time-consuming as it requires complex toll path planning. In this paper, a Jet-ECM variant is proposed which employs lithography before to fabricate surficial micro-features, which transforms Jet-ECM, the sequential process which fabricates microstructures one by one, to a parallel machining process, which numbers of microstructures can be fabricated at once. The processing accuracy depends strongly on the process voltage, mask resolution and nozzle speed. For microfabrication—in contrast with the traditional process, that is, milling and drilling, the free of tool wear cost make ECM based process economically; in contrast with Jet-ECM, the free of tool path design, the free of jet nozzle manufacturing and assembling, and the capacity to fabricate numerous microstructures at once make MEJM time-saving and effective; in contrast with TMEMM, the update and replacement of the electrolyte flow field constantly make batch fabrication with good reliability and repeatability. These pros offer the opportunity to fabricate complex surficial features with sizes of several microns on various metallic materials.

The present work demonstrated that the micro-letters of a patterned photoresist will replicate accurately (minimum shape error of 0.039%) on the workpiece surface at certain conditions. The excellent consistency of dimensional variation of 112 micro-features within per nozzle travel path indicated from SD (minimum SD of shape error ratio $\zeta_{s}:0.297\%$) demonstrates the reliability of MEJM. The experimental results suggest that a faster nozzle speed can improve the uniformity of microstructures in batch fabrication.

The conclusions of this work can be summarized as follows:1.A hybrid process of lithography and Jet-ECM is presented and the process capability is demonstrated through fabrication of complex microstructures such as micro letters, fishbone and concentric circle.
-quantitative assessment of machining depth and similarity ratio is presented in probability density function of distribution.-the probability density value of similarity ratio and machining depth both influenced by intergranular corrosion, and the value of the similarity ratio also susceptible to mask failure.2.surface structuring with good repeatability (minimum SD of shape error ratio $\zeta_{s}:0.297\%$) and shape accuracy (minimum shape error of 0.039%) is achieved.3.by analyzing the distribution of experimental results, faster nozzle speed suggests less mask failure possibility hence beneficial for improving shape accuracy. For applied voltage of *U* = 50 V, with the nozzle speed decreased from 2400 μm/s to 800 μm/s, the probability density value of similarity ratio decreased by 71.917%, where as the value of machining depth decreased by 47.790%.4.by analyzing the standard deviation of experimental results, a faster nozzle speed can improve the uniformity of microstructures in batch fabrication. The mean value of SD of the similarity ratio $\zeta_{s}$ decreased from 2.008 to 0.297, and mean value of SD of depth decreased from 0.747 to 0.080 as nozzle speed increased from 800 μm/s to 2400 μm/s.

Further studies should consider the fundamental mechanism of MEJM with coupled multiphysical fields in terms of electrolyte flow, electro-migration, concentration polarization, and other aspects. Factors influencing the process, such as the inter-electrode gap, the tool size, the nozzle travel speed, and the electrolyte flow rate, should be investigated. For other complex shapes, consideration should also be given to the use of a pulsed power supply and more sophisticated photoresist patterns. Strategies for determining the nozzle travel path when machining larger surfaces should also be the focus of future studies.

In conclusion, our results suggest a promising technology for fast and batch fabrication of surface microstructures and indicate its potential for industrial applications. 

## Figures and Tables

**Figure 1 micromachines-11-00948-f001:**
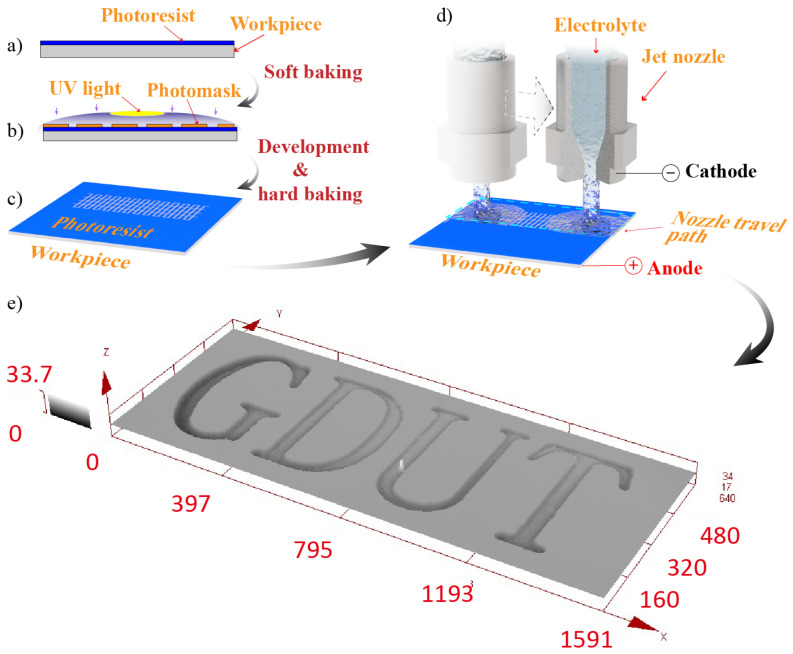
Schematic view of the steps in mask electrolyte jet machining process: (**a**) Photoresist coating; (**b**) Exposure; (**c**) Patterned photoresist; (**d**) Electrolyte jet mask machining; (**e**) Micro channels.

**Figure 2 micromachines-11-00948-f002:**
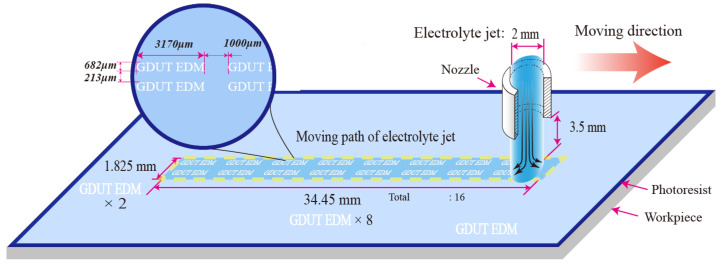
Schematic of experimental set up and pattern definition of MEJM.

**Figure 3 micromachines-11-00948-f003:**
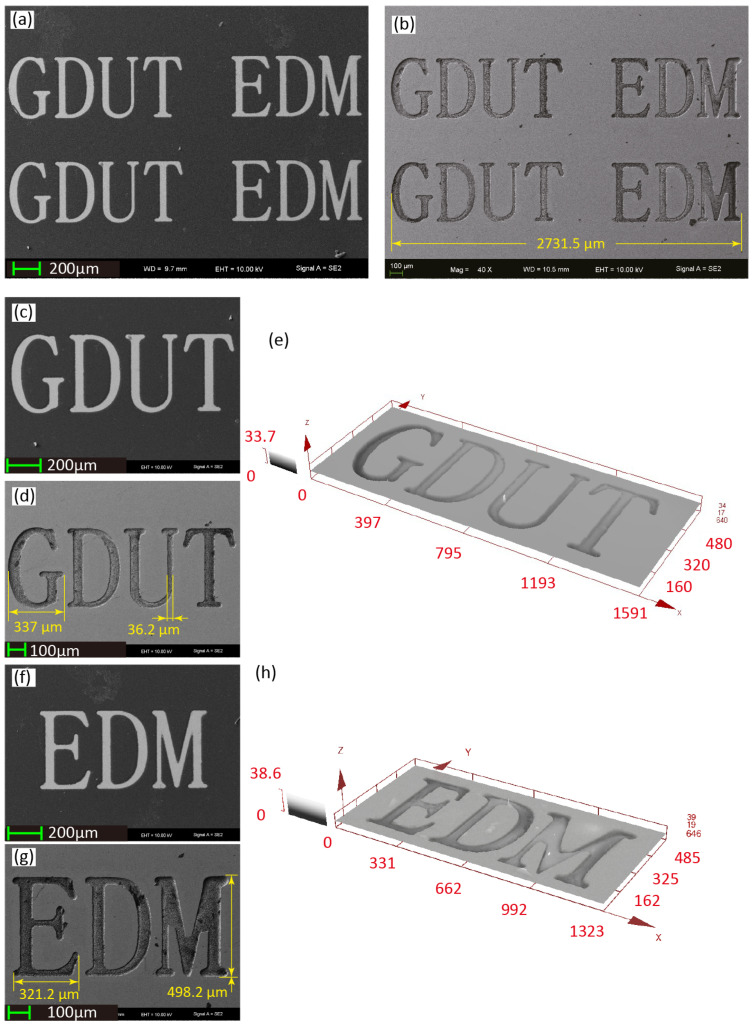
Patterned photoresist and corresponding micro features of fabricated by MEJM. (**a**) Two sets of micro letters of photoresist; (**b**)Two sets of micro letters fabricated by MEJM; (**c**) Micro letters “GDUT” of photoresist; (**d**) Micro letters “GDUT” fabricated by MEJM; (**e**) 3D profile of “GDUT”; (**f**) Micro letters “EDM” of photoresist; (**g**) Micro letters “EDM” fabricated by MEJM; (**h**) 3D profile of “EDM”.

**Figure 4 micromachines-11-00948-f004:**
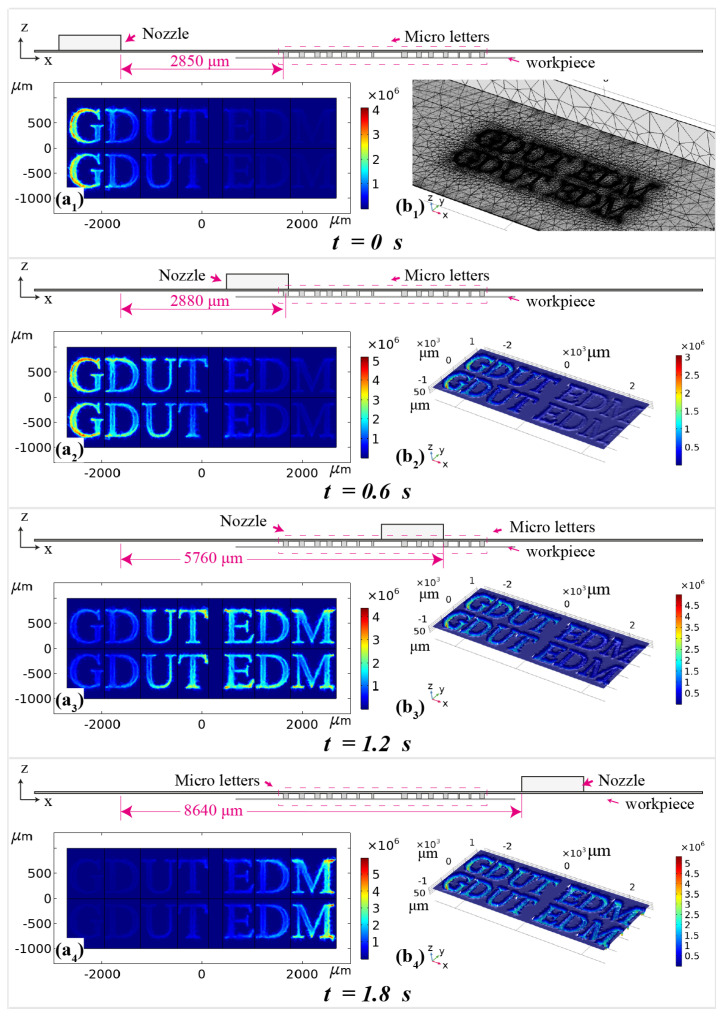
(**a${}_{1}$**–**a${}_{4}$**) Current density distribution (surface: *A*/m${}^{2}$), (**b${}_{1}$**) Mesh at *t* = 0 s, (**b${}_{2}$**–**b${}_{4}$**) Geometry shape depth (surface: $10^{-3}$ nm).

**Figure 5 micromachines-11-00948-f005:**
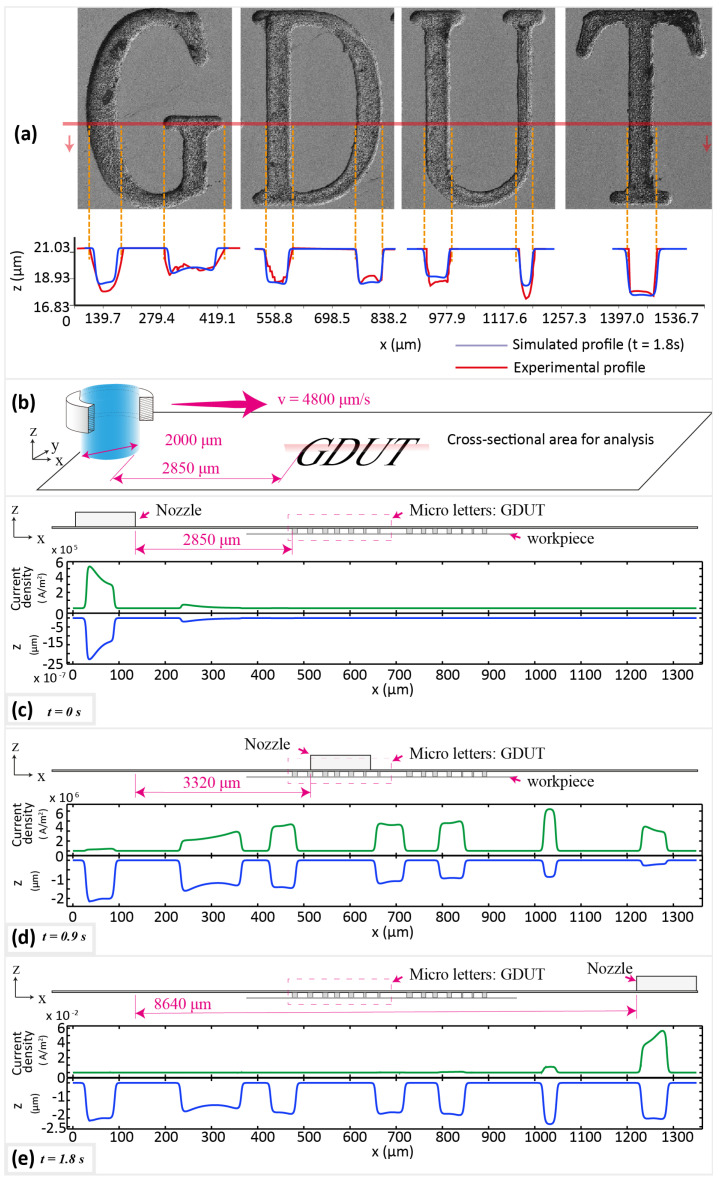
The effect of the topography of microstructures on the distribution of current density. (**a**) Simulated and experimental cross-sectional profile; (**b**) Schematic view of the cross-sectional research area; (**c**–**e**) Cross-sectional current density distribution and profile at *t* = 0 s, 0.9 s, 1.8 s.

**Figure 6 micromachines-11-00948-f006:**
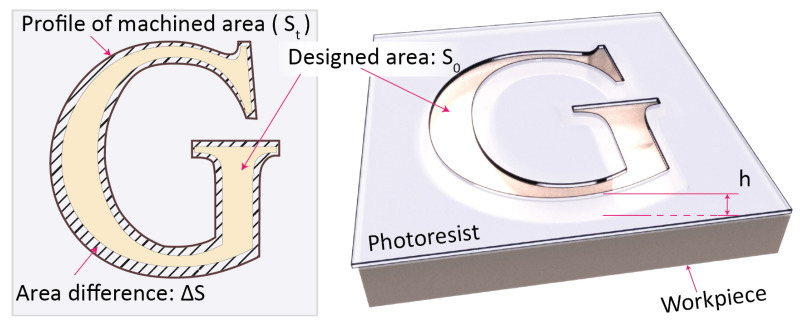
Basic characters of micro letters fabricated by MEJM.

**Figure 7 micromachines-11-00948-f007:**
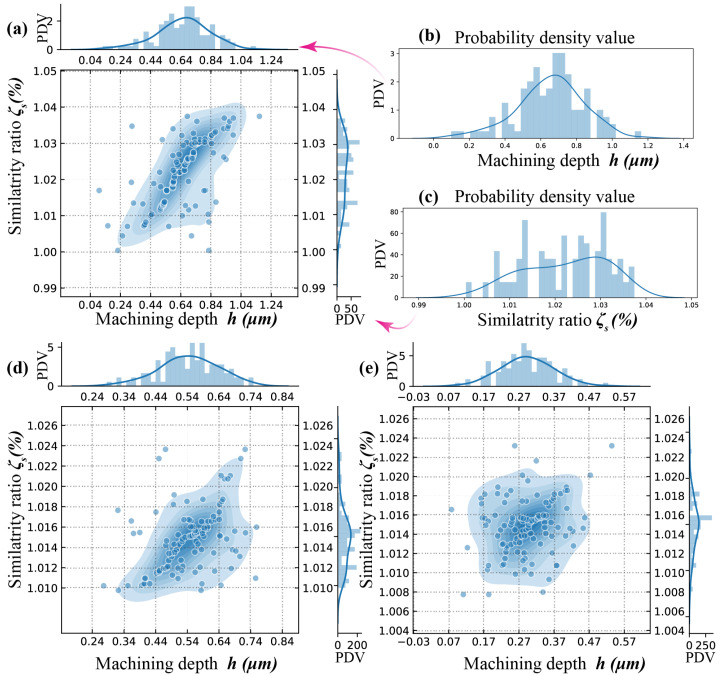
Distribution of similarity ratio $\zeta_{s}$ and machining depth *h* with different nozzle speed at applied voltage U=50 V: (**a**) $v_{1}$ = 800 μm/s; (**b**) distribution of machining depth *h*; (**c**) Distribution of similarity ratio $\zeta_{s}$; (**d**) $v_{2}$ = 2400 μm/s; (**e**) $v_{3}$ = 4800 μm/s.

**Figure 8 micromachines-11-00948-f008:**
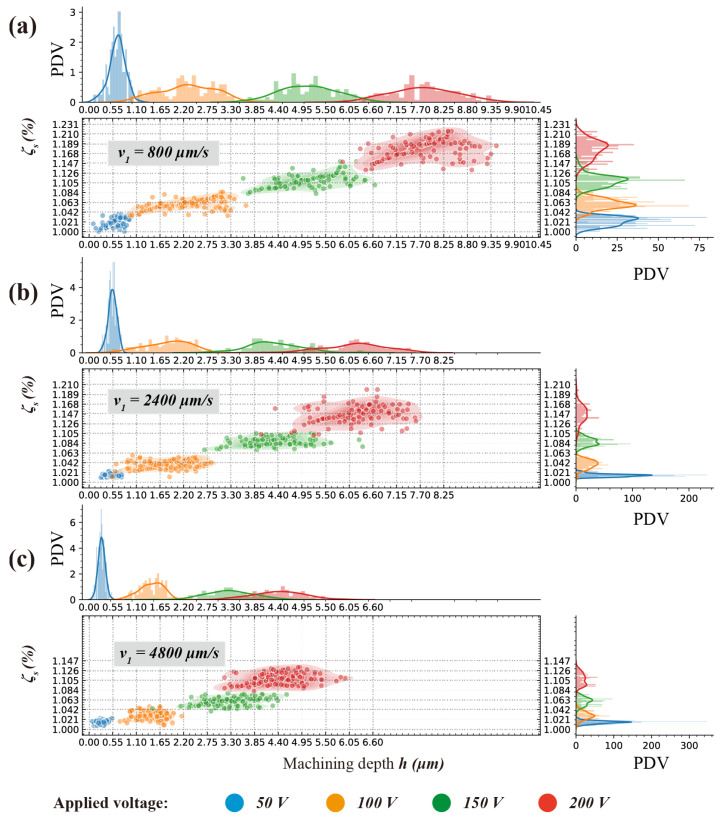
Effect of applied voltage *U* and nozzle travel speed *v* on the similarity ratio $\zeta_{s}$ and machining depth *h* of micro letters. (**a**) $v_{1}$ = 800 μm/s; (**b**) $v_{2}$ = 2400 μm/s; (**c**) $v_{3}$ = 4800 μm/s.

**Figure 9 micromachines-11-00948-f009:**
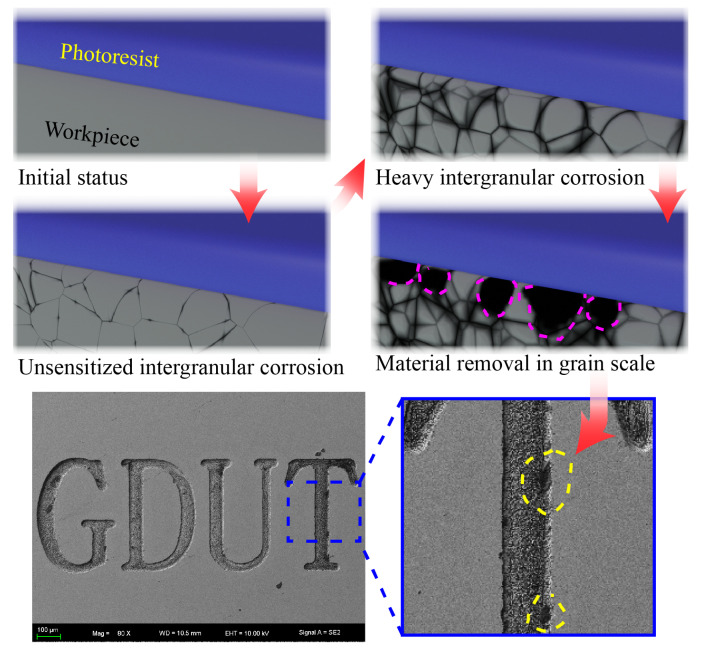
Illustration of intergranular corrosion in MEJM and its effect on fabricated shape.

**Figure 10 micromachines-11-00948-f010:**
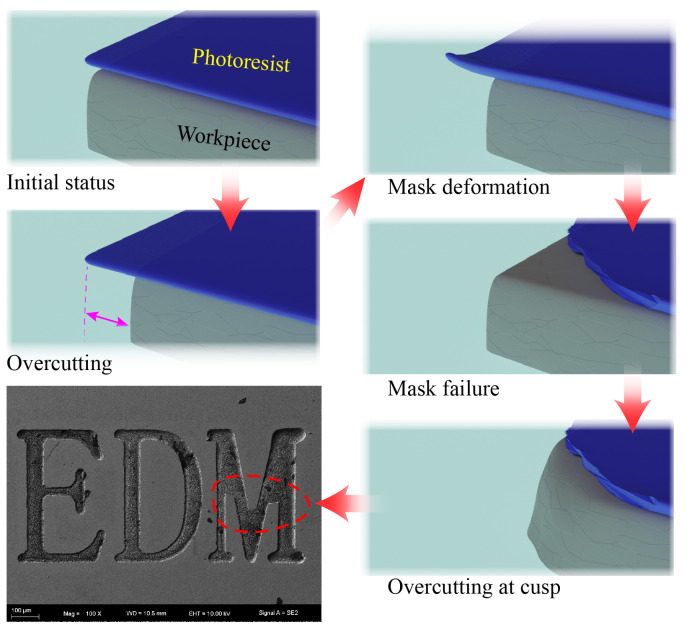
Illustration of mask failure in MEJM and corresponding effect on fabricated shape.

**Figure 11 micromachines-11-00948-f011:**
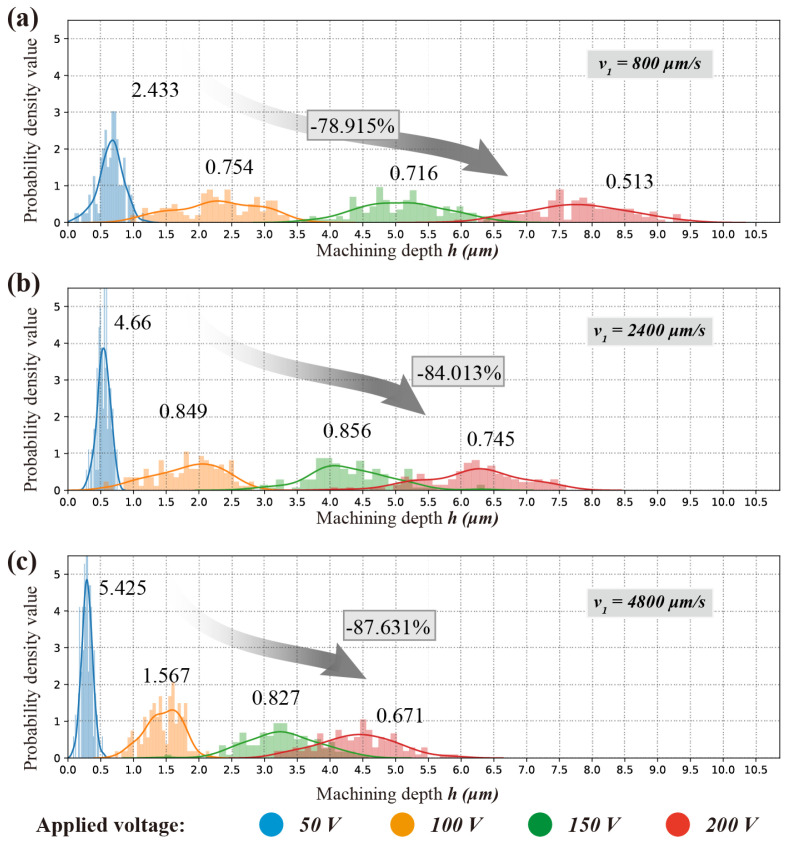
Probability density value of machining depth *h* with different nozzle speed: (**a**) $v_{1}$ = 800 μm/s; (**b**) $v_{2}$ = 2400 μm/s; (**c**) $v_{3}$ = 4800 μm/s.

**Figure 12 micromachines-11-00948-f012:**
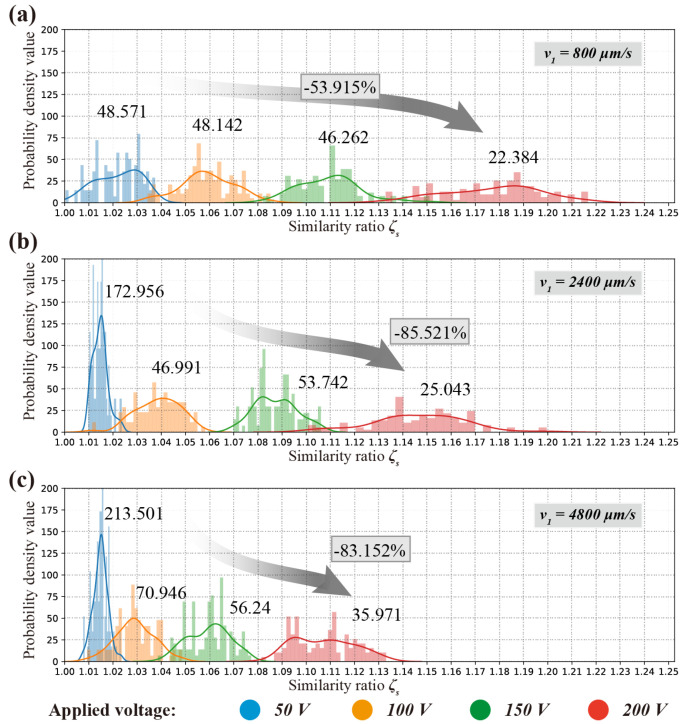
Probability density value of similarity ratio $\zeta_{s}$ with different nozzle speed: (**a**) $v_{1}$ = 800 μm/s; (**b**) $v_{2}$ = 2400 μm/s; (**c**) $v_{3}$ = 4800 μm/s.

**Figure 13 micromachines-11-00948-f013:**
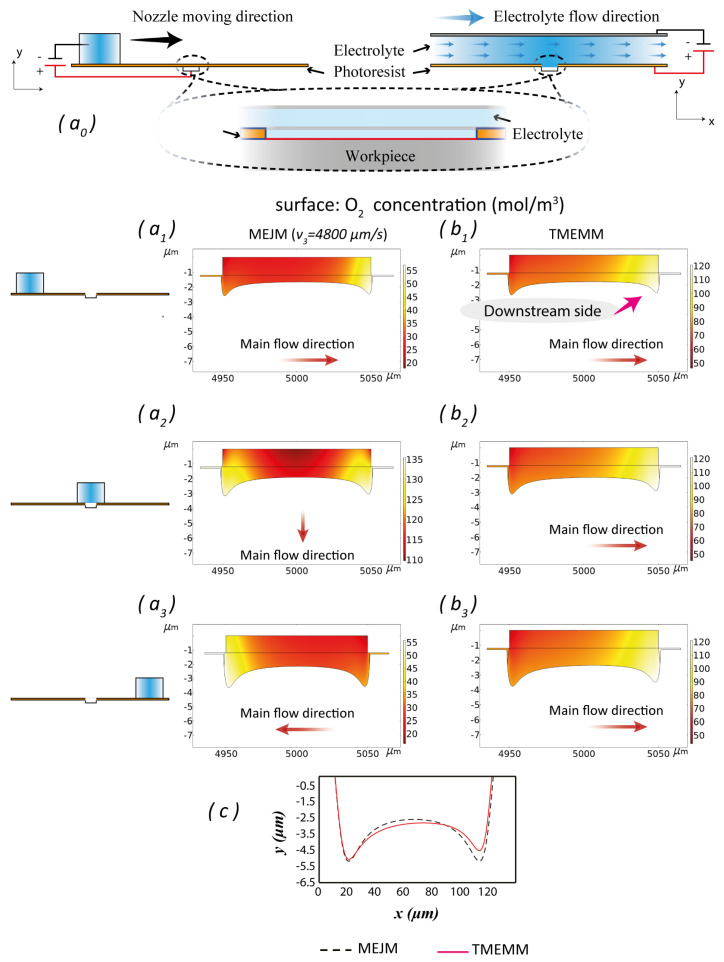
(**a${}_{0}$**) Schematic of simulations between MEJM and TMEMM; concentration of O${}_{2}$ in different environments: (**a${}_{1}$**–**a${}_{3}$**) MEJM with different nozzle positions; (**b${}_{1}$**–**b${}_{3}$**) TMEMM with the same maximum depth. (**c**) Comparison of simulated profiles of MEJM and TMEMM at the same maximum depth.

**Table 1 micromachines-11-00948-t001:** SD and $SD_{\mathrm{ratio}}$ comparison between mask electrolyte jet machining (MEJM) [25] and through mask electrochemical micro-machining (TMEMM) [13].

		MEJM [25]	TMEMM [13]
Width	SD	0.298	0.98
	$SD_{\mathrm{ratio}}$	0.029%	0.925%
Depth	SD	0.747	1.13
	$SD_{\mathrm{ratio}}$	9.603%	11.078%

**Table 2 micromachines-11-00948-t002:** Chemical composition of workpiece.

Composition	Content (wt%)
Cr	13.44
Mn	10.24
Fe	73.17
Ni	1.59
Cu	1.31

**Table 3 micromachines-11-00948-t003:** Experimental conditions for micro letters.

Parameters		Conditions
Workpiece material		Stainless steel
Electrolyte solution		NaNO${}_{3}$
Electrolyte concentration (mol·L${}^{-1}$)	*c*	0.8
Electrolyte pressure (kPa)	*P*	31
Nozzle inner diameter (mm)	$D_{nozzle}$	2
Thickness of the photoresist layer (μm)	$T_{p}$	1.3
Working gap (mm)	*d*	3.5
Applied voltage (V)	*U*	50, 100, 150, 200
Nozzle travel rate (μm·s${}^{-1}$)	*v*	800, 2400, 4800

**Table 4 micromachines-11-00948-t004:** Descriptive statistic of micro letter.

	*U*	Mean			Minimum			Maximum		
		$v_{1}$	$v_{2}$	$v_{3}$	$v_{1}$	$v_{2}$	$v_{3}$	$v_{1}$	$v_{2}$	$v_{3}$
	50	102.2	101.5	101.5	100.0	101.0	100.8	103.7	102.4	102.3
$\zeta_{s}$	100	105.9	103.9	102.9	103.4	101.1	100.9	108.5	105.8	104.9
(%)	150	111.0	108.7	106.0	108.2	107.0	103.9	115.1	110.6	107.7
	200	117.7	114.6	110.7	113.3	110.1	108.1	121.7	120.0	113.3
	50	2.210	1.478	1.488	0.039	0.977	0.774	3.745	2.364	2.319
$\zeta_{e}$	100	5.912	3.870	2.914	3.434	1.140	0.930	8.517	5.809	4.855
(%)	150	10.96	8.738	5.971	8.209	6.979	3.854	15.10	10.61	7.722
	200	17.70	14.64	10.73	13.29	10.07	8.149	21.66	19.95	13.31
	50	0.653	0.545	0.293	0.098	0.276	0.078	1.162	0.759	0.534
*h*	100	2.288	1.829	1.454	0.953	0.544	0.719	3.650	2.841	2.154
(μm)	150	5.098	4.298	3.275	3.584	2.683	1.475	6.646	6.361	4.594
	200	7.775	6.188	4.420	5.898	4.002	3.148	9.473	7.603	5.955

N total = 112.

**Table 5 micromachines-11-00948-t005:** Standard deviation of micro letter.

	*U*	SD		
		$v_{1}$	$v_{2}$	$v_{3}$
	50	0.937	0.309	0.297
$\zeta_{s}$	100	1.116	0.914	0.807
(%)	150	1.322	0.877	0.866
	200	2.008	1.925	1.239
	50	0.937	0.309	0.297
$\zeta_{e}$	100	1.116	0.914	0.807
(%)	150	1.322	0.877	0.866
	200	2.008	1.925	1.239
	50	0.186	0.097	0.080
*h*	100	0.625	0.521	0.274
(μm)	150	0.647	0.633	0.545
	200	0.747	0.726	0.589

N total = 112.

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
