# Peer review of "Fast Fabrication of Complex Surficial Micro-Features Using Sequential Lithography and Jet Electrochemical Machining"

_micromachines, 2020, doi:10.3390/mi11100948_

Round 1
Reviewer 1 Report
Please consider the comments and suggestions given in the commented version of your manuscript.

Reviewer 2 Report
Summary: The authors present an Electrochemical Machining process with the use of masks and jet electrolyte.
Comments
- The authors claim that this process can be used for batch manufacturing. How does it compare to traditional electrochemical etching performed after lithography? This needs to be added to the introduction section.
- While "hybrid" processes come with advantages they also carry the disadvantages of the respective processes that are combined. It is not clear how this process is better than the simple through mask ECM process without the jet. A comparison part with the same process parameters will add value to the paper.
- The authors claim this process does not require expensive equipment. Have they conducted a cost comparison study? This needs to be included to substantiate this claim.
- Was the simulation results in Figure 4 validated? Did the authors notice higher machining rates at the points with higher simulated current density?
- The authors claim that the simulated shapes were matching the experiments. This comparison must be presented in the paper.
Round 2
Reviewer 1 Report
Thank you very much for considering the comments and for cearfully revising the manuscript.
Reviewer 2 Report
Accept
This manuscript is a resubmission of an earlier submission. The following is a list of the peer review reports and author responses from that submission.
Round 1
Reviewer 1 Report
- Comments concerning introduction: in ECM not only saline solution are used, the statement that ECM is environmentally friendly alternative to CE is risky – I suggest to provide more details; Authors should clarify what means that jet-ECM is “sequential machining process”
- The research methodology should be improved: some discussion about process parameters should be included; table 1 with material composition is not necessary (is enough to give information about type with its name); in table 2 factor: “machining time” is unclear, is better to include path length; more details about dissolved area measurement procedure should be also included; the definition of shape accuracy EF (eq. 4) should be described in much more detailed way (how engineer can interpret this parameter?)
- According to eq (1). he shape error (eq.1) is not percentage parameter, however in tables 3 and 4 it is as [%]
- Instead word “discharge” I suggest to use “remove” or “evacuate” (p7, line 148)
- The result discussion should also contain section about edge quality and flatness (shape of its cross-section) of dissolved area.
- In my opinion some comments about adapt the process to machining larger areas are necessary. In presented case jet diameter is 2 mm. How will looks the process when machining i.e. area 10 mm x 10 mm ? What with the quality on the borders between jet passes? How such technology should be designed?
Reviewer 2 Report
Authors shall describe more precisely the method of producing elements for Jet ECM machining – chapter 2.1 (description of methods). When reading the paper, the process of producing the photoresist layer is not clear. It would be useful if steps were described more clearly. There are also missing parameters of the photoresist layer (like material, time) and description of the layer production process (hard baking, soft baking etc.).
Page 3, line 90 (section Experimental Setup) – authors write that “Fig. 2 shows the experimental system which consists of a mechanical system that provides movements to the electrolyte jet nozzle and workpiece for a suitable working space” there is probably figure / picture of the system missing in the paper, as the content and description of figure 2 is more suitable to what was written later on in page 4, line 111 (“Fig. 3 shows the photolithographic process to prepare the patterned photoresist, consisting of 112 micro letters, 16 sets of 7 letters; then micro letters were fabricated within every traveling path of the electrolyte jet...” – please clear the description of figures with figures content and further reference to the figures.
In Table 2, page 4, there are missing information important for the ECM process, concerning the electrolyte: temperature of the electrolyte (it would be useful to include also conductivity if the electrolyte) and electrolyte flow rate.
Table 2, page 4 – please explain parameter “machining time”? according to the scheme from fig. 2, ECM machining time shall vary, depending on the applied travel speed.
Page 4, Figure 3 – “Basic characters of micro letters fabricated by MEJM” – the current figure is a little misleading. When looking at the figure without big magnification, the right (pink) side looks like protrusion instead of groove or cavity. I would suggest to add cross-section of the machined shape for more clearance.
In the section 3.2 is missing the profile of the machined letters – 3D profile of the letter would be beneficial to analyse the variation of the letter depth depending on the machining area (it is slightly visible in the fig 4f, that near the letter border, the depth is different than in the middle). The 3 D profile of the machined letter will be used for evaluation of the current density along the machining gap during the Jet ECM process.
When analysing the ECM machining results authors do not mention current and current density during machining process, which is crucial for ECM efficiency – if it was measured, please consider adding results of current measurement during Jet ECM process.
In the discussion of the results authors concentrate on the averaged deviations in shape of the letters. It would be interesting and beneficial for potential readers, if authors presented discussion on the results of specific letter, taking into account the various dimensions – like in the letter “U” on the presented pattern. If possible, please include variations in the wider and narrower part of the letter. Is the deviation similar or it differs?
In table 2, page 4 (and fig. 2), authors indicate the distance between the nozzle and workpiece is set at 3,5mm – why is it set t this level? How will the change in working gap influence the machining accuracy and machining efficiency.
Paper needs proof reading towards improvement of:
- Spelling (i.e.: missing letters – page 4, line 119- missing “r” in the word “However”);
- Missing references (i.e.: page 7, line 148)
Reviewer 3 Report
This paper concerns the fabrication of micro-letters by masked EJM. In my opinion, the novelty is limited and there is a clear similarity to a paper, published recently, that is much more thorough in mechanistic understanding: doi.org/10.1016/j.ijmachtools.2019.103471. It is unclear where the novelty arises, beyond the shape of the mask.
In fact, some of the conclusions in this paper, e.g.: ‘Further studies should consider the fundamental mechanism of MEJM with coupled multiphysical fields in terms of electrolyte flow, electro-migration, concentration polarization, and other aspects. Factors influencing the process, such as the inter-electrode gap, the tool size, the nozzle travel speed, and the electrolyte flow rate, should be investigated.’, actually appear to have been investigated in this prior work.
The use of tables to present results (e.g. Tables 3-4) decreases the readability of what are some of the most important results in the paper. It is suggested that the authors should consider how to represent this graphically if possible.
Perhaps the paper would benefit from a mechanistic figure comparing the different flow conditions between TMEMM and MEJM.
Some of the statements in the literature review part of the introduction are confusing or inaccurate:
- ‘ECM uses a neutral saline solution. The electrolyte usually has high conductivity, low toxicity, and corrosive nature’. I think should be ‘non-corrosive’.
- ‘Minghuan Wang et al. [16] introduced the air-shielding in the EJM process’. This has actually been reported >12 years ago: (http://publica.fraunhofer.de/eprints/urn_nbn_de_0011-n-748717.pdf) and has been applied extensively by EJM practitioners from different research groups since.